# Machinability Measurements in Milling and Recurrence Analysis of Thin-Walled Elements Made of Polymer Composites

**DOI:** 10.3390/ma16134825

**Published:** 2023-07-04

**Authors:** Krzysztof Ciecieląg

**Affiliations:** Department of Production Engineering, Faculty of Mechanical Engineering, Lublin University of Technology, 36 Nadbystrzycka, 20-618 Lublin, Poland; k.ciecielag@pollub.pl

**Keywords:** milling, glass-fiber-reinforced plastics, carbon-fiber-reinforced plastics, thin-walled elements, feed force, deformation, recurrence method

## Abstract

The milling of polymer composites is a process that ensures dimensional and shape accuracy and appropriate surface quality. The shaping of thin-walled elements is a challenge owing to their deformation. This article presents the results of milling polymer composites made of glass and carbon fibers saturated with epoxy resin. The milling of each material was conducted using different tools (tools with polycrystalline diamond inserts, physically coated carbide inserts with titanium nitride and uncoated carbide inserts) to show differences in feed force and deformation after the machining of individual thin-walled samples. In addition, the study used recurrence analysis to determine the most appropriate quantifications sensitive to changes occurring in milling different materials with the use of different tools. The study showed that the highest forces occurred in milling thin-walled carbon-fiber-reinforced plastics using uncoated tools and the highest feeds per revolution and cutting speeds. The use of a high feed per revolution (0.8 mm/rev) in carbon-fiber-reinforced plastics machining by uncoated tools resulted in a maximum feed force of 1185 N. A cutting speed of 400 m/min resulted in a force of 754 N. The largest permanent deformation occurred in the milling of glass-fiber-reinforced composite samples with uncoated tools. The permanent deformation value of this material was 0.88 mm. Low feed per revolution (0.1 mm/rev) resulted in permanent deformations of less than 0.30 mm for both types of materials. A change in feed per revolution had the most significant effect on the deformations of thin-walled polymer composites. The analysis of forces and deformation made it possible to conclude that high feed per revolution were not recommended in composite milling. In addition to the analysis of machining thin-walled composites, the novelty of this study was also the use of recurrence methods. Recurrence methods were used to determine the most appropriate quantifications. Determinism, averaged diagonal length and entropy have been shown to be suitable quantifications for determining the type of machined material and the tools used.

## 1. Introduction

Composite materials that are an alternative to aluminum, titanium and magnesium alloys have been used for several decades, and their share in shaping processes is significantly increasing compared to other materials [1,2,3]. The demand for polymer composites and their processing is increasing due to their specific and beneficial properties. In particular, the properties of low specific weight and high strength mean they are widely used in elements where material costs are not the main production criterion. In an early period of the use of composite materials, the focus was on the needs of the space, defense and aviation industries, which meant that the development of these materials was oriented at achieving specified properties, and not reducing material costs. The relatively high price of composite materials and their machining is balanced by their advantages, namely, in addition to low specific weight and high strength, they have high corrosion resistance, vibration damping ability, ease of forming any shape and good electroinsulation properties [1]. Unfortunately, composites also have a shortcoming, which is their temperature resistance. The long-term use of composites at high temperatures is unfavorable for them. The upper approximate continuous working temperature of composites depending on the type of matrix (which is a factor determining the thermal properties of a composite) is 400 °C for the polymer matrix, 580 °C for the metallic matrix and 1000 °C for the ceramic matrix [4].

Given a growing demand for understanding the phenomena related to composites, many studies have been conducted on the machining of these materials in recent years. These are mainly studies investigating the influence of cutting parameters, type of tool and composite material on cutting forces, roughness parameters and surface topography.

A literature review shows that the technological parameters of milling polymer composites with glass and carbon fibers are within the following limits: depth of cut 0.1–4 mm [5,6], cutting speed 20–250 m/min [7], and feed per tooth 0.01–0.5 mm/tooth [7,8,9,10]. According to some studies, cutting speeds of up to 500 m/min are also used [11]. The ranges of the parameters affect the effects of milling. By increasing the cutting speed in machining glass-fiber-reinforced plastics (GFRP), tool wear is significantly increased [12], but at the same time, surface roughness is improved [5,13]. In turn, an increase in feed causes the deterioration of surface quality [5,14]. The machining of carbon-fiber-reinforced plastics (CFRPs) is characterized by similar relationships to those observed in the machining of glass-fiber-reinforced plastics. GFRP machining causes surface roughness to increase as the cutting speed decreases while the feed speed increases [15,16]. An increase in feed also has a negative effect on cutting forces, causing them to increase [17,18]. Researchers have also shown that an increase in cutting speed, which is beneficial for surface quality, unfortunately also causes damage in the form of delamination and leads to higher processing temperature and cutting forces [19].

In addition to the technological parameters of milling, another important aspect is the appropriate selection of tools. Among the tools used for milling polymer composites, there are tools with a polycrystalline diamond (PCD) insert, tools with a chemically applied thin coating (CVD), tools with a physically applied coating (PVD) and uncoated carbide tools [1,20]. This selection depends on the type of composite material, expected machining results and tool costs. In order to achieve a surface roughness characterized by low parameters, tools with polycrystalline diamond inserts are most often used [21]. The cost of the tools with polycrystalline diamond inserts is significantly higher than that of the tools with a physically or chemically applied coating or even uncoated carbide tools. Due to the high cost of PVD, CVD and uncoated tools are also successfully used with satisfactory results [22].

Previous research works devoted to the milling of polymer composites also focus on the type of composite material. Composites are materials with a heterogeneous structure, in which the type and arrangement of fibers determine the properties and workability of the entire material. The arrangement of fibers mainly affects the direction of the load transfer. In terms of obtaining low roughness values, an arrangement of 0° to 30° to the direction of machining is preferred. Exceeding the 30° angle causes the bending of the fibers and, consequently, deterioration of the surface quality [20,23]. The machining process conducted at an angle of 90° to the direction of the fiber arrangement additionally causes their shear [20]. When cutting carbon fiber with a 90° fiber orientation, in addition to shearing, the matrix breaks and the resin bond is damaged [20].

Due to the fact that composite materials have also been used as thin-walled elements, this aspect is the subject of this research. They are a very good alternative to thin-walled structures made of aluminum and titanium alloys [24,25], where it is necessary to ensure high stiffness and strength with a relatively low specific weight. Previous simulations of milling thin-walled composite structures based on FEM analysis enabled the analysis of the milling process plan, taking into account the impact of machining parameters [26]. Numerical analyses and experimental studies were also conducted to investigate the load capacity of thin-walled elements made of polymer composites. Based on their results, it was possible to determine the critical load of the structure [27]. Numerical calculations are a tool that also allows you to determine the stability of thin-walled structures [28]. Research has also been conducted on thin-walled structure machining, but it largely concerned the machining of aluminum alloys. The research in this field focuses on the determination of technological parameters of machining. The most significantly negative impact on deformation and surface roughness was observed by an increase in feed [29], and the second most important parameter was cutting speed. A significant influence of the machining strategy and wall thickness of the processed material on its dimensional accuracy and surface roughness was also found in [30,31]. The machining of thin-walled elements made of aluminum alloys was also studied in [32], where deformation was determined. Deformation is directly related to the resulting stresses generated during machining [33]. In terms of deformation research, it was also shown that even a two-fold reduction in the thickness of a thin-walled element made of aluminum alloy could lead to a double increase in deformation after machining [30]. For machining thin-walled elements made of aluminum alloys, tools with polycrystalline diamond inserts or tools made of sintered carbides are used [34].

In the analysis of economic, industrial and machining phenomena, recurrence plots (RPs) and recurrence quantifications (RQA) are used as machining analysis methods in research on various types of materials [35,36]. The recurrence plot is based on the method of delayed coordinates and is created based on the analysis of the signal which can be a force recorded during machining [37]. The reconstructed lag vector x obtained by the lag method is described by Formula (1):x = (x_i_, x_i+d_, x_i+2d_, … , x_i+(m−1)d_).(1)

Vector reconstruction involves selecting an embedding delay *d*, embedding dimension *m* and threshold parameter *ε*. In the formula, x_i_ denotes the i-th coordinate in a given time course [38]. The reconstructed vector is used to create a recurrence plot. However, due to the fact that the plots only provide qualitative information, recurrence quantifications have been introduced [39,40,41]. They are an advanced tool for the analysis of non-linear signals [42]. Among the many recurrence quantifications, one can distinguish the recurrence rate (RR, percentage of darkened points), determinism (DET, percentage of recurrence points which create diagonal lines), laminarity (LAM, percentage rate of recurrence point which create vertical lines), trapping time (TT, average length of the vertical lines), averaged diagonal length (L, average length of the diagonal line), longest vertical line (V_max_, longest vertical line of the recurrence structure), length of longest diagonal line (L_max_, longest diagonal line of the recurrence structure), recurrence time 1st, 2nd (T1, T2, time distances of the recurrence points in the vertical direction), entropy (ENTR, probability distribution of the diagonal line lengths), recurrence period density entropy (RPDE, measure quantifying the extent of recurrences) and clustering coefficient (CC, the probability that two recurrence states are close) [41,43,44,45]. Recurrence quantifications can be analyzed using two main methods. The first one is based on the assumption that the threshold parameter ε is constant. The other method is that the recurrence rate (RR) is constant and that the threshold parameter ε is changed to ensure a constant value of RR [39].

The use of recurrence methods is associated with recording a signal that serves for further analyses. This method has been successfully employed in medicine [46,47,48] and materials engineering research [49,50]. Recurrence methods have also been used to assess rotor cracks [51] and to analyze engine operation [52]. In machining, recurrence methods have also been applied. Studies have shown that the RQA technique has great potential in detecting surface wear in cutters. RQA parameters such as entropy (ENTR), trapping time (TT) and laminarity (LAM) can be used to detect insert wear in face milling [53]. In terms of machining, recurrence methods have been used to detect defects during drilling and milling of polymer composites. Studies [43,54,55,56] used cutting force as an input signal. Recurrence analyses allowed us to demonstrate that there were recurrence quantifications enabling the identification of defects. It was shown that the use of recurrence methods in drilling and milling made it possible not only to detect a defect, but also to determine its size and location. Previous studies also showed that it was possible to determine recurrence quantifications such as laminarity, entropy and recurrence time for detecting defects in polymer composites [43,55,56]. Recurrence analysis was used to study the milling of thin-walled structures made of aluminum alloys. The input signal for analysis was the cutting force, thanks to which the dynamics of the milling process was determined [57].

This article presents the research on milling thin-walled composite structures, a problem which has not been widely studied and is still new. Two different thin-walled composite materials consisting of glass- and carbon-reinforced fibers were used for the study, and they were machined using three types of cutting tools. The novelty of this work is the use of recurrence analysis to describe the cutting process of thin-walled polymer composites. The aim of the research was to determine the recurrence quantification that would be the most sensitive to the type of machined material and tool. In addition, the deformation of composite thin-walled structures was examined depending on the composite material, cutting tool and technological parameters of milling. Previous research on the use of recurrence analysis for defect detection in machining produced satisfactory results.

## 2. Materials and Methods

The milling experiments were carried out on the AVIA-VMC 800 HS vertical machining center. The center is controlled by Heidenhain iTNC 530. A 3D Kistler dynamometer (type 9257B) was placed in the working space of the machine tool. A vice was mounted at the top of the dynamometer. Test samples were fixed in the jaws of the vice. Each sample was clamped in the vice in the same way. The clamp of the vice for each of the samples ensured a stable fixation. In order to ensure the accuracy and repeatability of the results, the length and location of the milled horizontal surface relative to the mounting was identical for each research. The signal measured with the dynamometer was the feed force in accordance with the X axis of the working space and the dynamometer. The signal was processed by the Kistler charge amplifier (type 5070), the data acquisition card Dynoware (type 5697A) and the Dynoware software (type 2825A). A measuring probe was used to measure deformations of thin-walled elements made of polymer composites after milling. Measurements of the maximum feed force values and deformation were carried out seven times. The measurement of deformation is a permanent change of the shape of the workpiece expressed in mm. The deformation after milling marked on the chart is a deviation from the “ideal” sample size. The study also involved using recurrence methods. A research methodology scheme is shown in Figure 1.

The samples were rectangular plates with the dimensions of 10 mm × 10 mm × 100 mm. The width of the 10 mm sample was selected to ensure reliable milling with the entire diameter of the cutter, which was 12 mm. The thickness of the sample was selected in order to obtain results from four trials of milling composites of different thicknesses, i.e., 10 mm, 8 mm, 6 mm and 4 mm. The length of the sample of 100 mm resulted from the need to fully insert the tool into the material (30 mm) and to ensure stable clamping in the vice. The samples were made of glass and carbon-fiber-reinforced plastics saturated with epoxy resin. The first type of material was a glass-fiber-reinforced plastics (GFRP) called HexPly 916G-7781. It consists of 42 layers of pre-impregnated fabrics with a thickness of 0.24 mm. The material consisting of glass fibers and epoxy resin contains 47.68% carbon, 7.55% silicon, 3.89% nitrogen, 30.87% oxygen, 4.19% aluminum and 4.36% calcium. Other amounts of elements do not exceed 1%. The other type of material was a carbon-fiber-reinforced plastics (CFRP) called HexPly AG193PW-3501, consisting of 33 layers of pre-impregnated fabrics with a thickness of 0.3 mm. The material consisting of carbon fibers and epoxy resin contains 81.85% carbon, 5.71% nitrogen, 10.52% oxygen and 1.23% sulfur. Other amounts of elements do not exceed 1%. In each material, individual prepregs were arranged alternately (0°–90° arrangement). After assembling and placing them in a vacuum package, the samples were subjected to polymerization. This process took place in an autoclave for 3 h, at a pressure of 0.6 MPa. The process was carried out at 120 °C for GFRP and at 180 °C for CFRP.

The machining of thin-walled polymer composites was carried out at a constant depth of 2 mm and variable feed per revolution and cutting speeds, the values of which are listed in Table 1. Technological parameters of milling were selected on the basis of a literature analysis. Each subsequent value of feed per revolution and cutting speed is twice as large as the previous one. Milling was conducted on composite samples with a thickness of 10 mm, 8 mm, 6 mm and 4 mm, over a length of 30 mm.

The milling process was carried out using three different cutting tools that are used in the processing of polymer composites. Cutting tools are selected on the basis of a literature analysis and own research. Diamond-coated tools, coated and uncoated carbide tools are used for the machining of polymer composites. The tools were folding milling cutters with a diameter of 12 mm, equipped with a body of type R217.69-1212.0-06-2AN on which were mounted two polycrystalline diamond inserts with the symbol XOEX060204FR PCD05 (denoted by PCD), two physically coated carbide inserts with titanium nitride TiN XOEX060204FR-E03 F40M (denoted by F40M) and two uncoated carbide inserts XOEX060204FR-E03 H15 (denoted by H15). Detailed information about the cutting tools is presented in Table 2.

## 3. Results and Discussion

The research on the milling of thin-walled polymer composites made it possible to measure the feed force in order to present its value depending on the variable technological parameters, processed material and tool type. Standard deviations are marked in the plots in Figure 2, Figure 3, Figure 4a,b and Figure 5a,b. Small values of the standard deviation indicate that the results are close to the average value. Figure 2 shows the effect of feed per revolution on the maximum values of the feed force F in milling thin-walled polymer composites with a thickness of 4 mm, made of glass-fiber-reinforced (GFRP) and carbon-fiber-reinforced (CFRP) plastics, using three types of tools (PCD, F40M and H15).

The results presented in Figure 2 show that for each case of feed per revolution, higher maximum feed forces were obtained in machining thin-walled CFRP composites. At low feed per revolution values, the differences between the maximum force values for individual materials using the same tools are small. For the smallest value of the tested feed per revolution, maximum force values in the range of 260–390 N were obtained. Smaller force values (260–265 N) refer to the milling of GFRP and CFRP using PCD tools, and the largest (390 N) were obtained during the milling of CFRP using tools with uncoated inserts. At higher feeds per revolution, the differences begin to increase when comparing the maximum values of the feed force obtained for individual types of materials and tools. The most noticeable change is when milling CFRP with uncoated tools. For a feed per revolution of 0.2 mm/rev, a value of 685 N was obtained, and for a feed per revolution of 0.4 mm/rev, this value has already increased to 769 N. The highest values of the feed force were obtained as a result of machining thin-walled carbon-fiber-reinforced plastics using the uncoated carbide tools at a feed per revolution of 0.8 mm/rev. For the CFRP material, the use of uncoated tools has a negative effect on the force which reaches almost 1200 N. For comparison, the use of the PCD or F40M (PVD) tools at a feed per revolution of 0.8 mm/rev resulted in a reduction in the maximum feed force value by 20%. The increase in feed per revolution is significant in machining thin-walled polymer composites because for 0.8 mm/rev, when compared to 0.1 mm/rev, there was a three-fold increase in forces when the machining process was conducted with the uncoated tools. To sum up, the use of the highest feed per revolution value resulted in force values of 692 N, 711 N and 920 N, respectively, during GFRP milling for PCD, F40M and H15 tools. During CFRP machining, 935 N (for PCD), 974 N (for F40M) and 1185N (for H15) were obtained. A moving tool with a higher feed per revolution must remove the same amount of workpiece in less time. The use of a higher feed per revolution causes an increase in the tool load, and thus an increase in the maximum feed force values. Higher maximum feed force was created when milling CFRP because it is a material with greater strength.

Milling was also performed with variable cutting speed. Figure 3 shows the impact of cutting speed on the force values in milling glass- and carbon-fiber-reinforced plastics. The milling process was conducted with three different tools on the sample with a thickness of 4 mm.

An analysis of the cutting speeds showed that in the machining of thin-walled polymer composites, the maximum values of the feed force increase. The clearest increase was observed in the machining of carbon-fiber-reinforced plastics. Compared to the cutting speed of 50 m/min, the use of a cutting speed above 100 m/min caused a clear increase in the feed force in CFRP machining. The highest feed force values were obtained when the machining of both types of materials was conducted with the uncoated carbide tools. For the lowest cutting speed of 50 m/min, the maximum feed force values of 280 N were obtained when milling GFRP using PCD and F40M tools. The use of H15 tools, i.e., uncoated sintered carbides for GFRP machining, resulted in an increase in force to 414 N. Low cutting speeds during CFRP machining resulted in the values of 372 N (for PCD), 332 N (for F40M and 334 N (for H15). The increase in cutting speed meant that in the case of CFRP machining, the values of the maximum feed force increased significantly. For the cutting speed of 100 m/min, the use of PCD tools for CFRP machining resulted in a force of 483 N, for F40M a force of 498 N, and for H15 the force was already 682 N. A further increase in the cutting speed to 200 m/min and 400 m/min caused the maximum values of the feed force to continue to increase. At the highest cutting speed tested during CFRP machining, the forces obtained were 543 N (for PCD), 636 N (for F40M) and 754 N (for H15). The increase in the maximum value of the feed force when machining thin-walled GFRP and CFRP materials with the increase in cutting speed can be explained by the high resistance of the workpiece. One explanation is that despite the increasing cutting speed, and thus the higher rotational speed of the tool, composite materials are difficult to machine.

In addition to cutting force, the deformation of thin-walled polymer composites was also analyzed. Figure 4a,b show the influence of feed per revolution and tool type on the maximum permanent deformation of the GFRP and CFRP samples with a thickness of 4 mm after milling.

Figure 4a shows the maximum deformation obtained with increasing feed per revolution in milling the GFRP samples. For the smallest feed per revolution of 0.1 mm/rev, the deformation when machining GFRP with PCD tools is 0.14 mm, and for F40M and H15 tools, the deformation is about 0.30 mm. An increase in feed per revolution causes an increase in deformations, which for the highest feed per revolution value are 0.44 mm (for PCD), 0.58 mm (for F40M) and 0.88 mm (for H15). The highest deformations occurred in milling with the uncoated tools. A clear increase in deformations can be observed for a feed per revolution value of 0.8 mm/rev. Figure 4b shows the effect of feed per revolution on permanent deformation after CFRP milling. In the case of deformation analysis after CFRP machining, the values of 0.04 mm (for PCD), 0.10 mm (for F40M) and 0.25 mm (for H15) were obtained for the smallest value of the tested feed per revolution. Increasing the feed per revolution value caused further permanent deformations after CFRP machining and led to dimensional changes of 0.31 mm (for PCD), 0.41 mm (for F40M) and 0.78 mm (for H15) for the highest feed per revolution value. The results clearly show an increase in permanent deformation with increasing feed per revolution, with the largest deformation observed for the machining process conducted with the uncoated tools. A comparison of the two types of tested materials shows that higher deformations occurred in the sample made of GFRP. The average value of permanent deformation for the GFRP samples is 10–30% higher than that obtained for the CFRP material under the same processing conditions. The feed per revolution parameter is important when machining thin-walled polymer composites. Greater deformations during milling of composite materials along with increasing the feed per revolution are caused by the resistance that the tool must resist. As a result of increasing feed per revolution, the material is partially bent, leaving a permanent deformation after machining.

The plots in Figure 5a,b show the effect of cutting speed and tool type on the maximum permanent deformation of the GFRP and CFRP materials after milling. The thickness of the sample was 4 mm, and the milling depth was 2 mm.

Based on the plots illustrating the impact of cutting speed on permanent deformation, it can be concluded that cutting speed is a less important parameter than feed per revolution. The use of the polycrystalline diamond insert tools and the tools with a physically applied coating for different cutting parameters did not show any clear dependencies. For both types of materials, the use of these tools did not cause significant differences in deformation. Low cutting speed values caused GFRP deformation at the level of 0.02 mm for PCD and about 0.40 mm for F40M and H15. For the lowest cutting speed tested, the permanent deformation of CFRP was 0.02 mm (for PCD), 0.03 mm (for F40M) and 0.23 mm (for H15). Only the use of the uncoated tools for the cutting speed of 400 m/min resulted in a significant increase in deformation for both materials. Smaller values of the feed force and greater deformations of GFRP can be explained by the fact that this material is characterized by lower strength.

Another new aspect of the work is the introduction of recurrence analysis to the study of thin-walled polymer composites. Figure 6a–k show the values of recurrence quantifications depending on the tool type for the samples with a thickness of 4 mm, made of glass and carbon-fiber-reinforced plastics. The analysis was carried out with constant values of the milling parameters: f = 0.2 mm/rev and cutting speed v_c_ = 100 m/min. The recurrence analysis was carried out with a constant recurrence rate RR. To ensure a constant RR value, the threshold parameter was set to ε = 0.1. In the analysis, the embedding dimension was equal to 5 and the embedding delay was equal to 2.

The recurrence quantifications such as DET, L, ENTR, LAM, TT, T1, T2 and CC can be used to identify the type of material. The choice of these indicators is associated with a clear difference between their values for a given type of used tool.

Based on the above recurrence quantifications plots, it can be concluded that the indicators suitable for identifying the type of tool in GFRP milling are DET, L, ENTR, T1 and CC. This conclusion is supported by a clear change in the values of these quantifications in the plots. For the CFRP machining analysis, DET, L, L_max_, ENTR, TT, T2 and RPDE can be used to identify the tool type. DET, L and ENTR can be used to identify the type of tool in machining two types of materials.

## 4. Conclusions

This study investigated the machinability of thin-walled polymer composites. The milling of two types of composite materials reinforced with glass and carbon fibers was conducted using three types of tools. These were as follows: milling cutters with polycrystalline diamond inserts, physically coated carbide inserts with titanium nitride and uncoated carbide inserts. Machining was carried out with variable feed per revolution and cutting speed parameters. The results showed that the use of high feeds per revolution and high cutting speeds in machining thin-walled polymer composites had a negative effect on the maximum values of the feed force. For each tool, the milling of CFRP produced higher maximum cutting forces than those observed for GFRP. Due to the force values, the feed per revolution had the greatest impact on the machining of thin-walled polymer composites. A three-fold increase in the maximum feed force was obtained by comparing feeds per revolution of 0.8 mm/rev and 0.1 mm/rev. Changing the cutting speed also increased the maximum feed force. However, the cutting speed was the second most important milling parameter. The research results obtained in this article are consistent with the results of the work of other researchers whose works were quoted in the introduction. The tendency is confirmed that increasing the feed per revolution and cutting speed negatively affects the cutting force. The analysis of deformations after milling showed that the feed per revolution affected the machining effects for both types of materials and for each of the tools used. An increase in feed per revolution caused deformations after milling. The analysis of the impact of the cutting speed did not show significant dependencies, but the use of the uncoated carbide tools was found to significantly increase deformations. The research results for samples with a thickness of 10 mm, 8 mm, 6 mm and 4 mm are analogous. This article shows the results for the smallest thickness of the tested sample.

The novelty of this study is that it used recurrence analysis to select appropriate quantifications for the identification of the type of workpiece and tool. The recurrence methods made it possible to determine that the most popular indicators of tool and material identification were DET, L and ENTR. The plots generated for these indicators showed a change in their values depending on the type of tool and workpiece. This change was determined by comparing the maximum values of the feed force. An analysis of the maximum feed force showed that the high values resulted from the use of the uncoated tools for machining CFRP. This relationship can also be seen in the plots of recurrence quantifications. The DET values increase with the use of a tool that causes higher maximum feed force and large deformations. The DET quantification is characterized by higher values when machining CFRP depending on the tool type. The L indicator also increases when the inferior tools are used. However, in the machining of CFRP, these values were lower than the values obtained for GFRP. A similar relationship to the L indicator can be seen when analyzing changes in the ENTR indicator. The course of the indicators in the plots is similar to the analysis of the maximum values of the feed force.

Future research will study the geometric structure of thin-walled composite material structure after milling. Also, indicators dependent on technological parameters of milling will be determined via recurrence analysis. The proposed methodology for the research of thin-walled composite structures can be successfully used in an industrial environment. Cutting force measurements require the use of a dynamometer whose placement in the working space does not interfere with industrial conditions.

## Figures and Tables

**Figure 1 materials-16-04825-f001:**
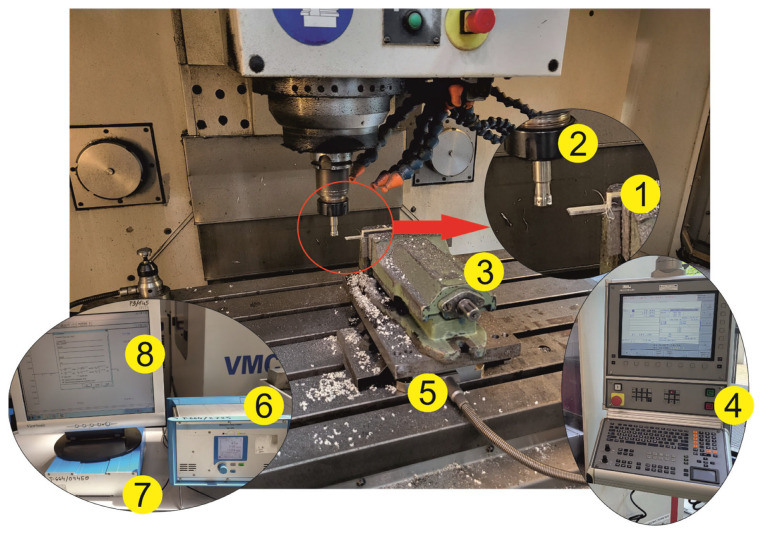
General scheme of the research methodology (1—sample; 2—tool; 3—vice; 4—control panel; 5—dynamometer; 6—charge amplifier; 7—data acquisition card; 8—software).

**Figure 2 materials-16-04825-f002:**
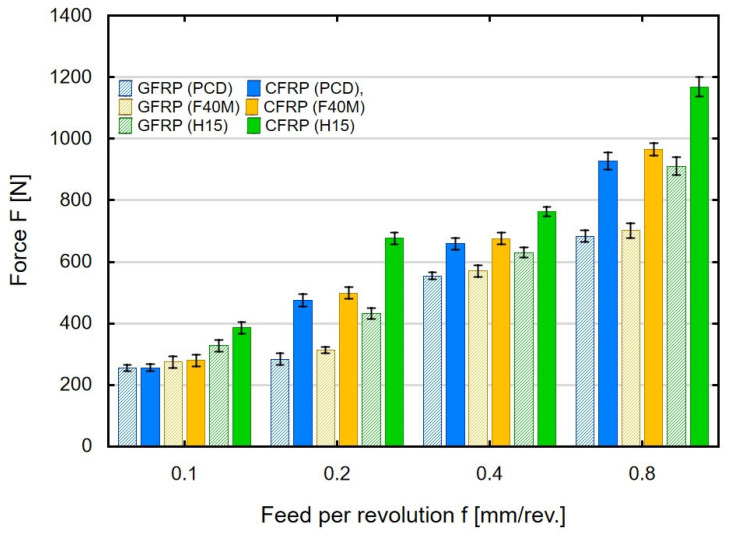
Feed per revolution versus maximum feed force in milling thin-walled polymer composites.

**Figure 3 materials-16-04825-f003:**
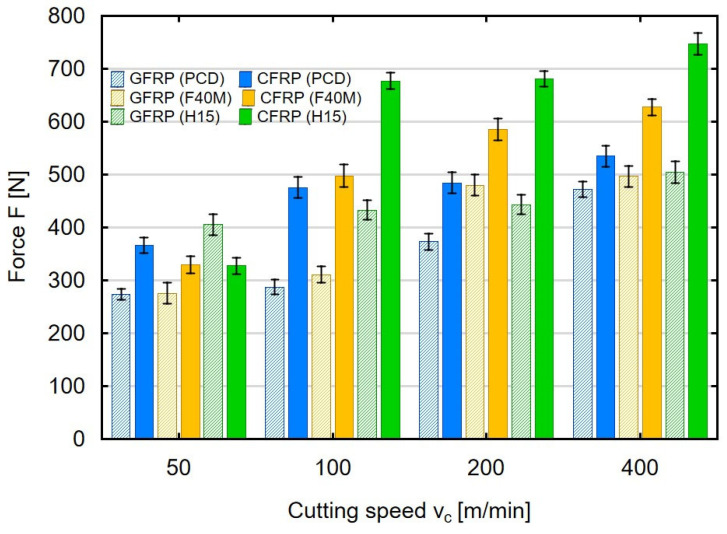
Cutting speed versus maximum feed force during milling of thin-walled polymer composites.

**Figure 4 materials-16-04825-f004:**
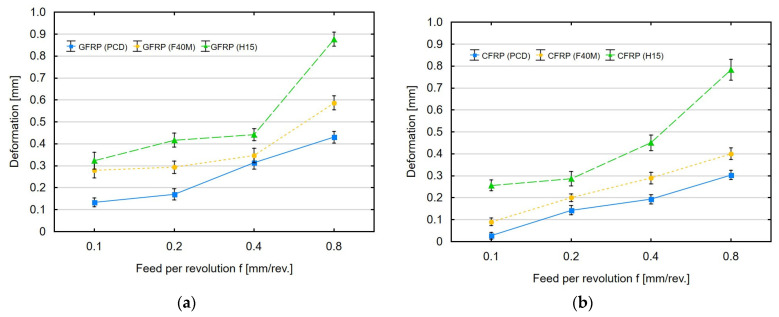
Feed per revolution versus permanent deformation after milling (**a**) GFRP and (**b**) CFRP.

**Figure 5 materials-16-04825-f005:**
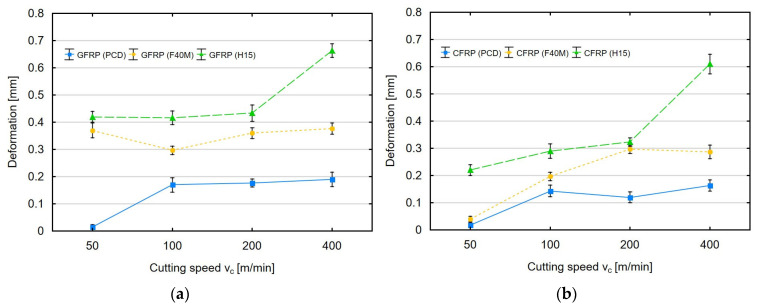
Cutting speed versus permanent deformation after milling for (**a**) GFRP and (**b**) CFRP.

**Figure 6 materials-16-04825-f006:**
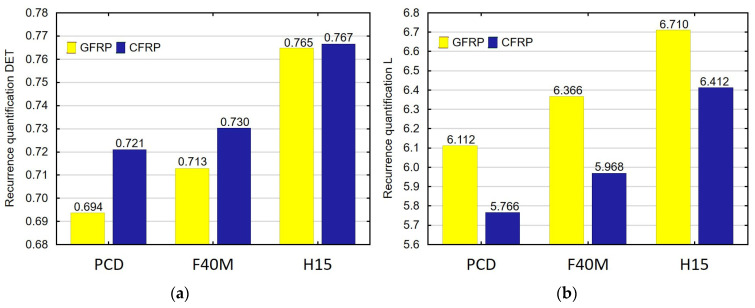
Type of tool and material versus recurrence quantifications: (**a**) DET, (**b**) L, (**c**) L_max_, (**d**) ENTR, (**e**) LAM, (**f**) TT, (**g**) V_max_, (**h**) T1, (**i**) T2, (**j**) RPDE and (**k**) CC.

**Table 1 materials-16-04825-t001:** Machining parameters.

No.	Feed per Revolution [mm/rev]	Cutting Speed v_c_ [m/min]	Depth of Cut a_p_ [mm]
1	0.1	100	2
2	0.2	100	2
3	0.4	100	2
4	0.8	100	2
5	0.2	50	2
6	0.2	100	2
7	0.2	200	2
8	0.2	400	2

**Table 2 materials-16-04825-t002:** Information about cutting tools (SECO).

Description	XOEX060204FR PCD05	XOEX060204FR-E03 F40M	XOEX060204FR-E03 H15
Gradetype	PCD	Carbide PVD	Carbide Uncoated
Clearance angle major	15°	15°	15°
Corner radius	0.4 mm	0.4 mm	0.4 mm
Cutting edge effective length	2.5 mm	6.0 mm	6.0 mm
Wiper edge length	1.1 mm	0.9 mm	0.9 mm
Insert thickness	2.45 mm	2.45 mm	2.45 mm

## Data Availability

Not applicable.

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
