# Peer review of "Machinability Measurements in Milling and Recurrence Analysis of Thin-Walled Elements Made of Polymer Composites"

_materials, 2023, doi:10.3390/ma16134825_

Round 1
Reviewer 1 Report
The manuscript presents the results of milling thin-walled composites made of glass and carbon fibers saturated with epoxy resin. After an intensive literature review, in the Introduction section, the scientific contribution and innovation of the proposed methodology are highlighted. The topic of the manuscript is original and relevant in the research field. The conclusions are consistent with the evidence presented. Selected references are appropriate. Figures and tables are clearly presented.
Suggestions for improving the manuscript are as follows:
1. No Type of the Paper is selected. You should select Article.
2. Exact numerical results could be stated in the abstract.
3. It should be mentioned the influence of fixture (vice) on the obtained results, the influence of locating accuracy, clamping force, compliance and the like.
4. In section 2, the materials and methods are listed, but the reasons are not explained. Why are the sample dimensions 10x10x100? How were the milling parameters selected (why exactly those values)? How are the cutting tools selected? Etc.
5. Detailed information about cutting tools should be displayed.
6. If possible, it would be good to show the chemical composition and mechanical, physical and thermal characteristics of the samples.
7. It is not clear how the experiment plan was set up. It should be elaborated. A full factorial experiment could have been done (although it is a more expensive variant).
8. Potential measurement errors and measurement uncertainty were not analysed and discussed.
9. The results are clearly presented and analysed. What the manuscript lacks is a more intensive discussion of the obtained results. The results should be discussed from the point of view of the physics of the process. Process discussion is missing from the manuscript. The influence of input variables on force, and explained in detail why they influence in that way.
10. Also, if possible, the results should be compared with the results of previous research.
11. If possible, it could be noted how the proposed methodology will be applied in a real industrial environment.
Author Response
Dear Reviewer,
I would like to thank the Editor for their consideration, and the Reviewer for the time spent on carefully reviewing this work and for their valuable deep insight and comments. I feel that this paper is now clearer, more thoroughly discussed and better-referenced.
Detailed answers to the questions have been added in the appendix.
Yours sincerely,
Krzysztof Ciecieląg

Author Response

(The authors gave the same response as above.)

Reviewer 3 Report
Good professional article, but with a low scientific level and with some fundamental shortcomings.
Measurement of deformation is not defined, parameter too.
Machinery technology is not well described (up milling, down milling?), feed is not well defined (must bee per tooth).
The article cannot be considered scientific.
The main question addressed by the research is deformation of composit material caused by milling.
The topic is relevant and not original in the proffesional field.
The paper is only proffesional, no scietific.
Measurement of deformation mast be fully defined, parameter too.Machinery technology is not well described.
The conclusions consistent with the evidence and arguments presented and do they address the main question posed but only on proffesional level, not scientific.
The references are appropriate.
The article cannot be considered scientific.
Author Response

(The authors gave the same response as above.)

Round 2
Reviewer 1 Report
The manuscript has been significantly corrected and supplemented. I suggest accepting the manuscript in its current form.
Reviewer 3 Report
All comments were sufficiently accepted, which significantly changed the value of the manuscript.